# Genome-Wide Classification and Phylogenetic Analyses of the GDSL-Type Esterase/Lipase (GELP) Family in Flowering Plants

**DOI:** 10.3390/ijms232012114

**Published:** 2022-10-11

**Authors:** Alberto Cenci, Mairenys Concepción-Hernández, Valentin Guignon, Geert Angenon, Mathieu Rouard

**Affiliations:** 1Bioversity International, Parc Scientifique Agropolis II, 34397 Montpellier, France; 2Instituto de Biotecnología de las Plantas, Universidad Central “Marta Abreu” de Las Villas (UCLV), Carretera a Camajuaní km 5.5, Santa Clara C.P. 54830, Villa Clara, Cuba; 3Research Group Plant Genetics, Vrije Universiteit Brussel (VUB), Pleinlaan 2, 1050 Brussels, Belgium

**Keywords:** gene families, GDSL-type esterase/lipase, plant genomes, orthology, database, bioinformatics, data curation, male sterility, pest and diseases

## Abstract

GDSL-type esterase/lipase (GELP) enzymes have key functions in plants, such as developmental processes, anther and pollen development, and responses to biotic and abiotic stresses. Genes that encode GELP belong to a complex and large gene family, ranging from tens to more than hundreds of members per plant species. To facilitate functional transfer between them, we conducted a genome-wide classification of GELP in 46 plant species. First, we applied an iterative phylogenetic method using a selected set of representative angiosperm genomes (three monocots and five dicots) and identified 10 main clusters, subdivided into 44 orthogroups (OGs). An expert curation for gene structures, orthogroup composition, and functional annotation was made based on a literature review. Then, using the HMM profiles as seeds, we expanded the classification to 46 plant species. Our results revealed the variable evolutionary dynamics between OGs in which some expanded, mostly through tandem duplications, while others were maintained as single copies. Among these, dicot-specific clusters and specific amplifications in monocots and wheat were characterized. This approach, by combining manual curation and automatic identification, was effective in characterizing a large gene family, allowing the establishment of a classification framework for gene function transfer and a better understanding of the evolutionary history of GELP.

## 1. Introduction

The first gene of the GDSL-type lipase/esterase (GELP) family was described by Upton and Buckley [1] in bacterial *Aeromonas* sp., in which the amino acid motif of the active site (GDSL) was distinguishable from that present in other lipases (GxSxG). Later, GELP genes were found to be widely distributed in all kingdoms. In plants, three main phylogenetic clusters have been observed [2] that have distinct exon/intron structures [2]. Through comparison of number and phylogenetic analysis of members of the GELP family, Volokita et al. [2] showed a general trend of gene expansion in higher plants, resulting in approximately 100 members in angiosperms.

GELP members are involved in a wide spectrum of functions, including hormone regulation, tissue development, and biotic and abiotic stress responses [3,4]. The possible reactivity of different substrates has been suggested for GELP enzymes [5]. Interest in the GELP family is growing and a number of crop-specific transcriptomic studies have been conducted in several plants, including *Brassica rapa* [6], *Sedum alfredii* [7], soybean [8], *Dasypyrum villosum* [9], tomato [10], and wheat [11], generating datasets at the genome scale. 

Currently, not many GELP genes have been functionally characterized, but it was recently shown that they can play an important role in anther and pollen development, and male sterility [11,12,13,14]. Therefore, this gene family has the potential to identify interesting candidate genes for crop improvement. 

However, transferring the knowledge acquired on model species to non-model species for such a large gene family requires homology identification. The identification of homologous relationships between genomes has been an active area of research, underpinned by the orthology conjecture that genes that diverge by speciation are functionally closer than those that diverged by duplication [15,16]. The relationship between two genes derived from a common ancestor by speciation is termed orthology. When several species are considered, the orthology relationship can be described by orthogroups (OGs), defined as the set of genes of a given sample of species that descended from a single ancestral gene that is present in the most recent common ancestor. Based on this definition, the determination of orthogroups is objective (based on phylogenetic relationships) and relative (depending on the species considered). 

As an evolutionary concept, homology is best inferred using phylogenetic analysis [17]. However, reconstructing phylogenies for large gene families can be challenging because of the high number of homologous genes, the presence of pseudogenes and possible annotation errors that introduce mistakes in sequence alignment and phylogeny inference. Although automatic methods continue to improve [18,19], automatic analyses have been shown to produce inaccurate results of homologous clustering in the large family of NAC transcription factors [20]. We showed that an expert-curated orthogroup framework among GELP genes is useful in obtaining the most accurate phylogenetic relationships among genes of large families [20,21,22].

In this study, we performed a semi-automatic curation of genes of the complex and large gene family of GELP, using multi-step phylogenetic analyses, to establish orthogroups relative to the ancestral split between monocot and dicots. We further explored their gene structures and functionally characterized GELP genes in the literature. Finally, we performed an automated genome-wide identification of 46 plant species to identify gene evolutionary dynamics and facilitate gene annotation transfer for functional studies. 

## 2. Results

All protein sequences annotated as GELP in nine angiosperm species that represent major angiosperm branches (855 sequences) were manually inspected and where necessary, gene structure corrections of the sequences were performed (Appendix A).

### 2.1. Iterative Phylogeny Analysis of GELP Sequences

The first global GELP phylogeny that included *A. trichopoda*, *P. dactylifera,* and *V. vinifera* (66, 81 and 101, respectively) resulted in one hundred and two positions after alignment filtering (Table 1), accounting for approximately one-fourth of the GELP sequence average size. In the unrooted phylogenetic tree, a cluster (Cluster 1) that contained 14 sequences (two, six and six, respectively, for the above-cited species) was clearly separated from all the other species (Figure 1; Appendix A). These sequences were removed from the dataset and the phylogenetic analysis was repeated.

In the second round of phylogenetic analysis, 113 positions were retained after the masking step and were used for gene tree reconstruction (Table 1). The phylogenetic tree was again characterized by a cluster (Cluster 2) that was well separated from the rest of the tree (Figure 1; Appendix A). Cluster 2 contained 12 sequences, 3, 5 and 4 for the species cited above. The process was repeated six more times (Table 1) until step 8, when the tree was split into three clusters (Clusters 8, 9, and 10) (Figure 1; Appendix A). In total, 10 well-supported clusters were identified. 

### 2.2. Orthogroup Definition

To establish monocot/dicot orthogroups (OGs) for the whole GELP family, 10 clusters, including the GELP sequences from *A. trichopoda*, 3 monocots, and 5 dicots were submitted for cluster-specific phylogenetic analysis. The number of sequences in the phylogenetic tree and remaining aligned positions are summarized in Table 2.

#### 2.2.1. Cluster 1

The phylogenetic tree displayed two main sub-clusters (Appendix A), each containing at least one sequence for each of the nine species. In one sub-cluster, dicot and monocot sequences (5 and 10, respectively) were grouped separately, and the *A. trichopoda* sequence showed a basal position, consistent with the known phylogenetic relationships. The second subcluster was less resolved. Considering only branch support of >0.9, we found species-specific sub-clusters, isolated sequences (including *A. trichopoda*), and a dicot and monocot sub-cluster. Based on these results, two orthogroups were established, OG-GELP-C1a and OG-GELP-C1b (Table 2 and Table 3). 

#### 2.2.2. Cluster 2

In this cluster, we identified three main sub-clusters (Appendix A) with aLRT support between 89 and 100, all containing at least one sequence for each of the analyzed species. Three OGs were established, OG-GELP-C2a, -C2b and -C2c (Table 2 and Table 3). 

#### 2.2.3. Cluster 3

Here, two main clusters, one containing monocot species and the other with dicots and *A. trichopoda*, were present (Appendix A). The sequences were included in OG-GELP-C3 (Table 2 and Table 3). Cluster 3 was characterized by the GDSY sequence that replaced the GDSL motif.

#### 2.2.4. Cluster 4

The phylogenetic analysis resulted in a tree with two large unresolved sub-clusters and three strongly supported sub-clusters (aLRT > 95); with the last three containing at least one sequence from each of the nine species and a unique *A. trichopoda* sequence in basal positions (Appendix A). For each of the last three subclusters, which are as follows, an orthogroup was established: OG-GELP-C4a, -C4b, and -C4c (Table 2 and Table 3). The two subclusters with poor resolution (hereafter called C4X and C4Y) were submitted separately for new phylogenetic analysis iterations.

##### Sub-Cluster C4X

The new analyses showed two well-supported clusters (Appendix A). In one, both monocot and dicot sequences were represented, as was a sequence from *A. trichopoda*. However, this cluster lacked clear resolution and the main nodes showed poor support. The other cluster contained a sub-cluster of four *A. trichopoda* sequences with basal positions and three highly supported clusters, containing at least one representative dicot sequence and no monocot sequences. Consequently, two orthogroups were established from this subcluster, OG-GELP-4e and 4f (Table 2 and Table 3). Interestingly, the sequences included in both clusters were physically close in four dicots, including in *A. trichopoda* (all five sequences), *P. mume* chromosome LG2 (LOC103321832/LOC103321833 and LOC103321831), *V. vinifera* chromosome 1 (LOC100264465/LOC100259265/LOC100264440 and LOC109122650), *T. cacao* chromosome 2 (LOC18609585/LOC18609586 and LOC18609588), and *C. canephora* chromosome 11 (CDP13237_Cc11_g15880 and CDP13238_Cc11_g15890). This suggests the occurrence of tandem duplications in the ancestors of the two clusters before the split between *A. trichopoda* and the ancestor of the monocots and dicots, followed by one copy lost in the monocot lineage, and triplication in the dicot lineage before the asterid/rosid split. 

##### Sub-Cluster 4Y

Four clusters were resolved by the tree (Appendix A). A cluster that contained 10 sequences from the nine species was sharply separated from the other sequences and was named OG-GELP-C4d. Additionally, all *A. trichopoda* sequences clustered together, and the other sequences formed three main clusters, including both the monocot and dicot species. Consequently, three additional OGs were established, OG-GELP-4g, C4h and C4i (Table 2 and Table 3). Copy number expansion was observed in both species and taxonomically wider lineages.

#### 2.2.5. Cluster 5

No *M. acuminata* or *O. sativa* sequences were found, making the two *P. dactylifera* sequences the only representatives of monocots in our panel of species. The three main clusters were well resolved. The first was *A. trichopoda*-specific, the second had two tandemly located sequences from both *A. trichopoda* and *P. dactylifera* and one included all dicot species (except *A. thaliana*). The third largest cluster included 43 sequences from all dicots structured in sub-clusters, some of which were species-specific and composed of several tandem duplicated sequences. Even if the phylogeny was not completely resolved, two orthogroups were established based on the second and third clusters, OG-GELP-C5a and C5b, respectively (Table 2 and Table 3). The sub-cluster with eight *A. trichopoda* sequences and six genes organized in tandem was considered closer to OG-GELP-C5b. 

#### 2.2.6. Cluster 6

Three well-supported sub-clusters (aLRT > 0.95) contained at least one representative sequence per species (Appendix A). Three orthogroups (OG-GELP-C6a, C6b, and C6c) were established based on these subclusters (Table 2 and Table 3).

#### 2.2.7. Cluster 7

The tree enabled the identification of eight OGs based on the main tree sub-clusters that contained at least one sequence of *A. trichopoda* and one from a monocot or dicot species (Appendix A). OG-GELP-C7a contained a sequence for each species considered in this study. OG-GELP-7f lacked monocots (Table 2 and Table 3) and OG-GELP-C7b was the largest OG in Cluster 7. In the monocot/dicot evolution, repeated duplications occurred after the split from the *A. trichopoda* lineage, increasing the number of copies in all the species. In OG-GELP-C7c, one sequence was included for each species except for *O. sativa*, which had two sequences, one with a basal position in the sub-cluster that contained the other sequences included in the OG. OG-GELP-C7d lacks sequences from *O. sativa*, and two *M. acuminata* sequences with isolated basal positions were tentatively included in the OG with poor support. OG-GELP-C7e contains only a few sequences that belong to both monocots and dicots, as well as *A. trichopoda*. In *V. vinifera* and *T. cacao*, the genes underwent gene copy amplification. Only sequences from the dicot species were included in OG-GELP-C7f. An *A. trichopoda* sequence (AMBTC_LOC18433946) appears close to this OG; however, aLRT support was very low and its association with the OG-GELP-C7f should be considered uncertain. OG-GELP-C7g contains sequences from all species except *A. thaliana*. Similarity analyses indicated that all species in the Brassicales order are missing members of the OG-GELP-C7g. One sequence from *A. trichopoda* and one from *V. vinifera* were associated with a two-sequence highly supported cluster. BLASTp analyses in the non-redundant protein database found only one sequence of *Nelumbo nucifera* (XP_010258913) that was close to the one in this cluster. According to the phylogenetic evidence, the OG-GELP-C7h was established including only these two sequences.

#### 2.2.8. Cluster 8

Two orthogroups, OG-GELP-C8a and C8b, were established according to two strongly supported clusters (aLRT = 0.99) (Appendix A), both containing sequences from monocots and dicots and one or two sequences of *A. trichopoda* (Table 2 and Table 3). The 84 sequences contained in the remaining poorly resolved Cluster C8X were used to build a tree based on 275 positions. This tree was structured into two main clusters (aLRT = 0.99), called C8X1 (29 sequences) and C8X2 (55 sequences) (Appendix A). For each sub-cluster, phylogenetic analysis was iterated. The tree obtained using the C8X1 sequences was based on 295 positions (Appendix A). The following three OGs were defined using these sequences (Table 2 and Table 3): OG-GELP-C8c, with high branch support, missing monocot sequences and with an *A. trichopoda* sequence; two other OGs were established in clusters supported at 0.90 (OG-GELP-C8d) and 0.96 (OG-GELP-C8e), both containing monocot and dicot sequences. An *A. trichopoda* sequence showed an intermediate position (basal to the OG-GDSM-C8d cluster but with a support lower than 0.9). The tree obtained with the 55 C8X2 sequences (279 positions) was structured into two large clusters, one specific to monocots and the other specific to dicots (Figure 2; Appendix A). Consequently, all these sequences were included in a unique OG, named OG-GELP-C8f (Table 2 and Table 3). In this OG, copy amplification occurred, increasing the gene copy number. In dicots, the amplification appeared to be more recent than the radiation of the analyzed species (21 *A. thaliana* sequences are present, organized in 3 main species-specific clusters, whereas 7 sequences are in a *V. vinifera*-specific cluster). In contrast, in monocots, gene amplification took place before the radiation of Commelinids (which includes the three sampled monocot species), generating at least four ancestral genes still present in the sample species (Figure 2).

#### 2.2.9. Cluster 9

The sequences were distributed in two main clusters, containing sequences from monocots, dicots, and *A. trichopoda* (Appendix A). Two OGs were established, OG-GELP-C9a and OG-GELP-C9b (Table 2 and Table 3). In the second cluster, dicot-specific duplication can be inferred from the tree.

#### 2.2.10. Cluster 10

Eight well-supported clusters (aLRT > 0.95) were distinguished (Appendix A). Six of them were small (three genes or less) but contained at least one sequence of *A. trichopoda* and representative sequences of both monocots and dicots. Therefore, six OGs were established (OG-GELP-C10a to OG-GELP-C10f in Table 3).

Two additional large subclusters (C10X and C10Y) contained a higher number of sequences per species. They were poorly resolved and were not organized into two clearly separated monocot- and dicot-specific subclusters. By building a new phylogenetic tree with C10X, two main clusters were observed (aLRT = 0.91), one containing all monocot sequences and the other including one *A. trichopoda* and dicot sequences (Appendix A). The phylogeny of the C10Y subcluster could not be completely resolved but two main clusters were observed, one with all monocots and the other with most dicots (Appendix A). The clusters C10X and C10Y were used to create OG-GELP-C10g and OG-GELP-C10h, respectively (Table 3). In OG-GELP-C10g, gene copy amplification before dicot species radiation was observed, whereas in monocots, gene copy amplification was mainly lineage-specific (Figure 2). 

### 2.3. Gene Structure of GELP Genes

#### 2.3.1. Exon–Intron Architecture

All 44 OG showed a consistent exon structure, with a few exceptions. When the exon structures were compared among OGs, the following five main structures were observed (Table 3): (1) sequences in OGs derived from Cluster 1 had only two exons with aligned exon junctions, with the second exon including more than 85% of the gene length; (2) sequences from Cluster 2 (OG-GELP-C2a, -C2b and -C2c) had similar structures with aligned exon junctions. Sequences in OG-GELP-C2a and C2b had six exons, and OG-GELP-C2c had five, due to the fusion of the second and third exons; (3) sequences in OG-GELP-C3 had five exons with unique junction positions compared to other OGs; (4) sequences in eight of the nine OGs derived from Cluster 4 had five exons with aligned junctions; the remaining one, OG-GELP-C4d, was composed of only two exons, obtained by the fusion of three and two exons, respectively; (5) sequences in all other OGs shared a basic five-exon structure, similar to the one observed for the OGs derived from Cluster 4; however, the junction between exons 2 and 3 is different. 

Overall, we observed that the number of exons could be reduced by the fusion of two or more exons or increased by the insertion of an intron in exon 3 (OG-GELP-C10e). Some dicot sequences (OG-GELP-C4a) had two or three alternative first exons (Table 3).

#### 2.3.2. Functional Domain Structure

With the exception of Clusters 1 and 2, the protein signature domain analysis of the 855 protein sequences identified the SGNH hydrolase superfamily domain (IPR036514) that included the GDSL lipase/esterase domains (IPR001087/IPR035669). These sequences also contained a signal peptide in the first 20–30 amino acids of the proteins. Cluster 1 and Cluster 2 displayed an alpha/beta hydrolase fold domain (IPR029058) and an SGNH hydrolase superfamily domain (IPR036514), respectively (Appendix A). 

### 2.4. Functionally Characterized GELP Genes

Several GELP genes have been functionally characterized in the literature. These experimentally characterized genes were assigned to specific orthogroups using a similarity search between the sequences in our sampling (Table 4, Appendix A). The main reported functions have been associated with anther and pollen development, male fertility, acetylcholinesterase activity, and biotic and abiotic stresses. 

### 2.5. Automatic Genome-Wide Classification 

The set of nine species of GELP sequences (Appendix A) was used as a verified kernel for orthogroup inference of the GELP family on a larger panel of plant genomes, which was composed of 46 species. (Appendix A). The GELP genes ranged from 59 in *A. trichopoda* to 346 in the polyploid genome of *Triticum turgidum,* with a median value of 120 genes. When normalized with the total number of the genomes, the GELP represented an average of 0.3%, with the least abundant in *Olea europea* and the most abundant in *T. turgidum* (Figure 3). A few inconsistencies in GELP gene numbers were observed with the species in the curated dataset, mainly due to differences in annotation. The only inaccurate OG assignment was for *A. trichopoda* assigned to OG-GELP-C9b, which was added to OG-GELP-C9a. However, because *A. trichopoda* genes were not included in the OGs (based on the monocot/dicot split), they were not considered for HMM pattern definition.

As previously detected in the core set, five dicot-specific orthogroups (OG-GELP-C4f, C5b, C7f, C8c and C10c) were confirmed. Orthogroups with expansion in monocots were also observed (OG-GELP-C4h, OG-GELP-C4i and OG-GELP-C10f). 

A phylogenetic analysis specifically computed in OG-GELP-C3 to compare member genes in the *Triticum turgidum* genome and *Brachypodium distachyon* showed large lineage amplification of wheat genes on two main loci, both on chromosome 7B (for the A genome, one on chromosome 4 probably because of the known 4A/7B translocation). Twelve wheat genes were clustered in a well-supported cluster (aLRT = 0.99), with two *B. distachyon* genes (Bradi3g35300.2 and Bradi4g28226.1) as sister taxa (Figure 4).

## 3. Discussion

### 3.1. An Iterative Phylogenetic Approach for Large Gene Families

Reconstruction of phylogenetic relationships among genes that belong to families with large numbers of members is challenged by the number of sequences. The calculation time increases exponentially with the number of sequences and sequence divergence can make the alignment process more problematic. Furthermore, gene sequence inaccuracies produced during the automatic annotation process can introduce biases in alignments [46]. To minimize these issues, sequence accuracy can be improved by inspecting sequences, particularly when parts appear poorly aligned to the closest homologs. In our study, genomic sequences were manually inspected, and when needed, alternative annotations were manually determined based on their similarity to homologs. Pseudogene sequences were identified if evidence of mutations was found, and were bypassed by automatic annotation by exon/intron structure adjustment.

To cope with the difficulty caused by the large numbers of sequences, we adopted strategies to reduce the number of analyzed sequences. Nine species (with more than 850 GELP sequences) were initially selected to represent the major angiosperm branches. The sequences from three species were used in global phylogenetic reconstruction to determine major clusters, including one that represented monocots (*P. dactylifera*), one that represented dicots (*V. vinifera*), and *A. trichopoda*, which has a phylogenetic position basal to the monocot/dicot lineage split. The representative species for the monocot and dicot lineages were chosen based on the lowest number of whole-genome duplications in their genome history. Then, the sequences of the six other species were integrated into ten clusters (C1 to C10) to perform the definitive phylogenetic analysis and determine the OGs.

To reduce the difficulties induced by the sequence number in alignment and phylogenetic accuracy, we applied an iterative approach. In fact, the larger the number of sequences analyzed, the lower the number of positions for phylogenetic inference after removing poorly aligned columns. A clear negative correlation between the number of sequences analyzed and aligned positions can be inferred from Table 1 and Table 2. Phylogenetic trees were cleaved into main clusters with very strong support and the phylogenetic process (alignment, masking and phylogenetic inference) was repeated for each reduced dataset. This zoomed-in approach allows the recovery of informative positions discarded in more complex alignments and provides better phylogenetic resolution. 

Eight analysis iterations on the three sampled species (248 sequences of the whole genomic GELP set) resulted in ten clusters (Figure 1). Except for a few gene-specific peculiarities, the exon/intron structures were consistent with this first-level classification (Table 3). Clusters 1, 2, and 3 had completely independent intron positions. The exon structures of sequences in Clusters 4 and 5–10 were different from those in the former clusters and were partially consistent, with the only difference being the insertion point of the second intron.

### 3.2. Comparison with Previous Published Studies

The iterative phylogenetic reconstruction of the GELP family obtained in this study was consistent with previous phylogenetic reconstructions based on the complete set of GELP genes in *O. sativa*, *A. thaliana*, and *P. mume* [3,47,48,49]. However, the sequence set was not uniform among studies. For example, the sequences that composed Cluster 1 were not considered in any of them. Although these sequences have been annotated as GDSL esterase/lipases in several species because they share some degree of homology with other sequences, they encode proteins that do not bear the GDSL motif. They were annotated as alpha/beta-hydrolases in *A. thaliana*. Consequently, the separated clustering of these sequences—with only two exons—was not surprising. In addition, sequences in Cluster 2 were not considered by Volokita et al. [3] or Lai et al. [48] and even if counted among rice GELP genes by Chepyshko et al. [47], the four *O. sativa* sequences in Cluster 2 were excluded from the phylogenetic analysis. In the study by Cao et al. [49] on Rosaceae, the *P. mume* genes included in Cluster 2 are part of Subfamily A. Cluster 2 sequences have GSSI or GDSI sequences instead of the typical GDSL.

Sequences in Cluster 3 were characterized by the GDSY sequence at the GDSL motif. This cluster included the sequences in Clade C in the work of Volokita et al. [3], Clade IVa described by Lai et al. [48], Clade II by Chepyshko et al. [47], and Subfamily C by Cao et al. [49]. 

The fourth phylogenetic iteration clearly separated sequences in Cluster 4 that were included in Clade B by Volokita et al. [3], Clades IVb and IVc by Lai et al. [48], Clade I by Chepyshko et al. [47], and subfamily D, as described by Cao et al. [49].

Finally, the Clades denoted A1 and A2 in the work of Volokita et al. [3], as well as Clades denoted I- II and Clade III in the work of Lai et al. [48], correspond to our Clusters 6–10 and 5, respectively. The sequences of *O. sativa* in Clades III and IV in the work of Chepyshko et al. [47] correspond to Clusters 5–10. The *P. mume* sequences included in Subfamilies B and E-J of the Rosaceae GELP [49] are represented in Clusters 5–10. 

A few inconsistencies with previously inferred phylogenies were observed in the lower rank (more recent clades) of our phylogeny and we consider that the larger species sample, along with the lower sequence divergence in our iterated phylogenetic analyses, provides a more accurate phylogenetic inference and granular classification of the GELP gene family. Based on the monocot/dicot split, 44 OGs were established, most of which contained representatives of all the species (Table 3). 

### 3.3. Extending Classification to New Genomes

Our approach of automatically extending seed orthogroups with a large panel of full genomes proved to be effective in representing the entire monocot and dicot phyla. This probably resulted from the careful selection of core datasets composed of three monocots and five dicots. To estimate the relevance of the reduced species sampling, we compared the obtained automated classification based on the total gene number of the 44 OG-GELP. The correlation was higher than 95%, suggesting that the genome selected to establish the core sequences was not biased in favor of some OGs (Figure 5). For the species present in both curated and automatically annotated sets, small discrepancies in the sequence number assigned to OGs were due to differences in annotation (pseudogene/remnant with sufficiently long sequences to be captured by the HMM profile in most cases). The results obtained with the HMM profiles may suffer from some inaccuracies in species outside the monocot/dicot clades (e.g., Gymnosperms).

### 3.4. Gene Duplicates

Copy number amplification appears to be a global trend in the evolution of the GELP family; however, our results indicated that in some OGs, the copy number increased after the monocot/dicot split; in some others, even phylogenetically close OGs, the copy number did not change and was limited to one sequence per species (Table 3 and Appendix A). The copy number expansion was due to gene duplications, both local (tandem duplication) and distal (possibly generated by whole-genome duplications). The observed duplications were either lineage-specific or had taken place before species radiation (e.g., OG-GELP-C8f and C4f, Figure 2). Extreme cases of GELP gene amplification were present in the 21 *A. thaliana* genes in OG-GELP-C8f. These differences suggest that some GELP ancestor genes evolved under different evolutionary forces, possibly because of their different functions. For each OG, the detailed duplication history is presented in the respective phylogenetic trees (Appendix A). Tandem duplications contributed to gene copy amplification after the monocot/dicot split, but also before this evolutionary milestone, as indicated by the tandem position of representatives of OG-GELP-C8a and C8c, in *A. trichopoda*, *C. canephora*, *A. thaliana* and *T. cacao* or of OG-GELP-C4e and C4f in *A. trichopoda*, *P. mume*, *V. vinifera*, *C. canephora* and *T. cacao*.

### 3.5. Functional Transfer in Large Gene Families

All the genes included in the OGs were derived from the same ancestor gene that was present in the common ancestor species of monocots and dicots. The OG framework should facilitate the transfer of functional annotation between species, highlighting the phylogenetically closest genes. The OG framework inferred in the present study provides detailed information that allows discrimination between gene lineages that underwent diversification and those that appear to be stable from an evolutionary point of view. Notably, several GELP genes were found to be involved in male sterility (Table 4). These genes were classified into several OGs (OG-GELP-C7a, C8f, C10a, C10b, and C10h), suggesting that different stages of pollen development require reactions that involve lipid molecules. Researchers interested in inducing male sterility in a given plant species should target GELP that belongs to OG-GELP-C7a, C10a or C10b, for which one or a very few copies per species are consistently observed, compared to OG-GELP-8f or C10h, whose ancestors underwent copy amplification. The functional transfer of genes that belong to OGs with a high number of members should be carefully evaluated. In fact, the persistence of multiple copies derived from a single ancestor gene could be the consequence of subfunctionalization or neofunctionalization that occurred among phylogenetically close genes, and independently evolved new functions are likely to be different. For instance, in GELP, we observed phylogenetic proximity between genes involved in apparently different functions, as well as phylogenetic distance between genes involved in similar functions. An example of this is the *A. thaliana* gene FXG1 (α-fucosidase) [50], which belongs to the same OG-GELP-C4e as the *Medicago sativa* gene that codes for the ENOD8 protein (present in root nodules) [51], the *Daucus carota* gene that codes for the EP4 glycoprotein observed in the cell wall of seedling roots [25] and the genes that code for *Zea mays* and *A. thaliana* acetylcholine esterase [24,52]. Conversely, the acetylcholine esterase genes isolated in *Macroptilium atropurpureum* [27] and *Salicornia europaea* [28] belong to the phylogenetically close but different OG-GELP-C4f, which also includes the *Gossypium hirsutum* gene GhGLIP involved in seed development [26].

## 4. Materials and Methods

### 4.1. Sequence Retrieval and Curation

All sequences annotated as “GDSL esterase/lipase” in *Amborella trichopoda* (GCA_000471905.1) [53], *Phoenix dactylifera* (GCA_000413155.1) [54], *Musa acuminata* v2 (GCA_904845865.1) [55], *Oryza sativa* (GCA_001433935.1), *Coffea canephora* v1 (GCA_900059795.1), *Vitis vinifera* (GCA_000003745.2) [56], *Prunus mume* v1 (GCA_000346735.1) [57], *Theobroma cacao* v2 (GCF_000208745.1) [58], and *Arabidopsis thaliana* (GCF_000001735.4) [59] were retrieved from GenBank. BLASTp analysis [60] was used to search for misannotated genes using the *V. vinifera* GDSL esterase/lipase genes as a query in the database of non-redundant protein sequences (nr) of the nine sampled species. Sequences were visually inspected and, when necessary, corrections were made to structural annotation (modified sequences are marked by ‘*’). Truncated sequences (missing part of the coding region due to genome sequence gaps) and probable pseudogenes with well-conserved protein sequences were tagged but maintained in the analyses. 

### 4.2. Phylogenetic Analyses

To reduce the complexity of the analyses and maintain angiosperm representation, 248 sequences from a subset of species were used to investigate the global GELP phylogeny, which were as follows: *A. trichopoda*, *P. dactylifera,* and *V. vinifera*. Protein sequences were aligned with the MAFFT program (https://mafft.cbrc.jp (accessed on 6 October 2022)) [61], using the EMBL-EBI bioinformatics interface [62] with default parameters. Sequence alignments were cleaned using GBlocks V0.91b [63]. Cleaning was performed by allowing (i) smaller final blocks, (ii) gap positions within the final blocks, and (iii) less strict flanking positions. Phylogenetic analyses were performed with PhyML [64] (available at http://www.phylogeny.fr (accessed on 6 October 2022) [65]), using an LG substitution model and an approximate likelihood-ratio test (aLRT). The unrooted phylogenetic trees were visualized using MEGA6 [66].

### 4.3. Orthogroups Curation

The orthogroups (OGs) were established using phylogenetic trees, including nine species, and identified based on the last common ancestor of monocots and dicots (three and five species, respectively). *A. trichopoda,* as an outgroup to the monocot/dicot ancestral node, was not included in the OGs. The OGs were delineated according to the smallest clusters, including GELP sequences from both monocots and dicots. All sequence compositions of the OGs were scanned with InterProScan v5.41-78.0 [67].

### 4.4. Literature Curation

The classification of GELP genes was performed using BLASTp in a database that contained GELP genes in this study. For each sequence that did not belong to the nine species studied, the assignment was made if at least the first five hits consistently belonged to the same OG. The list by Ding et al. [4] was combined with some other GELP genes.

### 4.5. Automatic Sequence Assignation to Curated OGs

Multiple alignments of the 44 OGs were used to generate HMM profiles with hmmbuild from the HMMer v3.2 software [68]. We developed a Perl script (https://github.com/guignonv/search_fasta_hmm (accessed on 6 October 2022)) based on HMMsearch with e-value 10e-3 to identify GELP sequences in the genomes of 46 species from the GreenPhyl database [69]. 

## Figures and Tables

**Figure 1 ijms-23-12114-f001:**
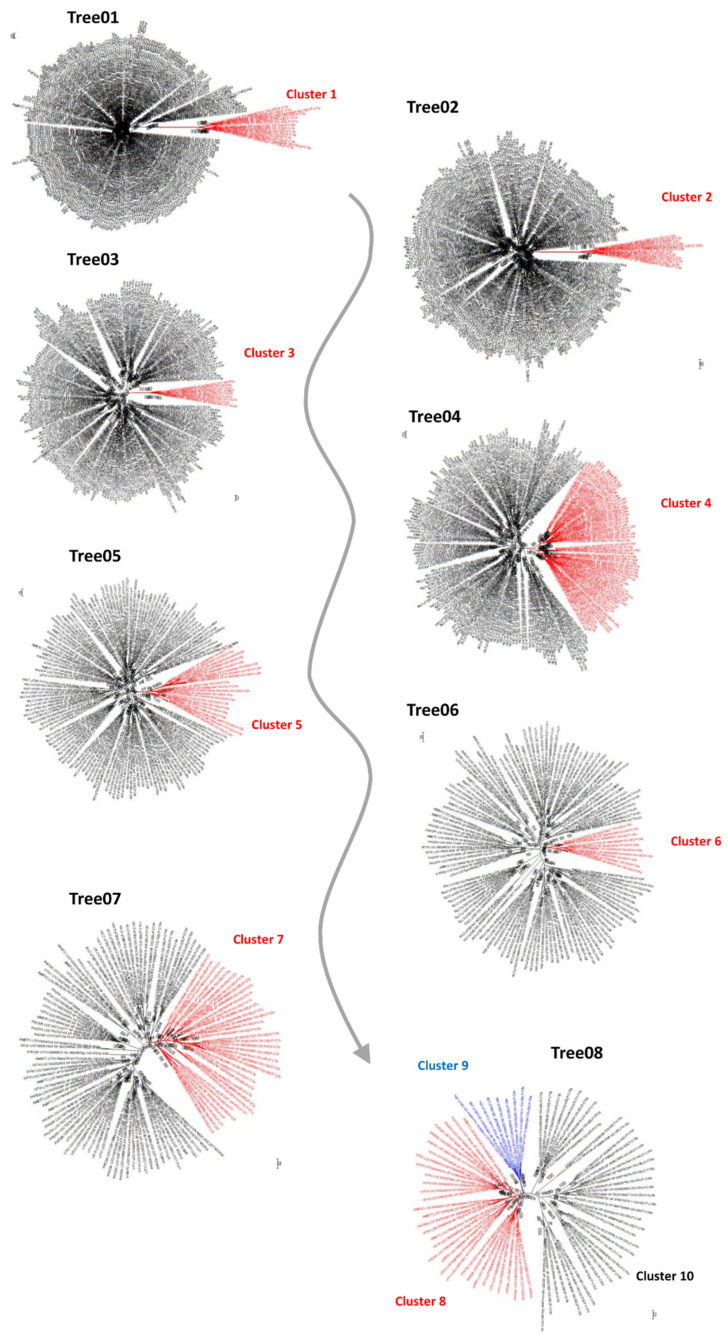
Schematic illustration of the iterative method using unrooted phylogenetic trees to identify the main clusters.

**Figure 2 ijms-23-12114-f002:**
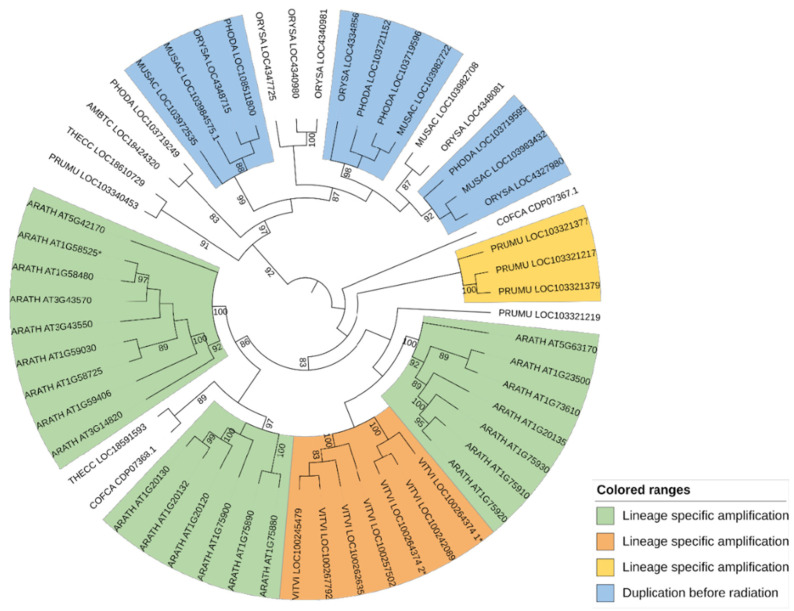
Phylogenetic tree in circular mode for OG-GELP-C8f rooted on node separating monocots and dicots. Colored subclusters indicate genes that originated either from duplications that occurred before species radiation (in monocots) or species-specific duplication (in dicots). Branch support with aLRT support lower than 0.8 is not displayed. * denotes curated sequences compared to automatic annotation (1* and 2* for split chimeric sequences).

**Figure 3 ijms-23-12114-f003:**
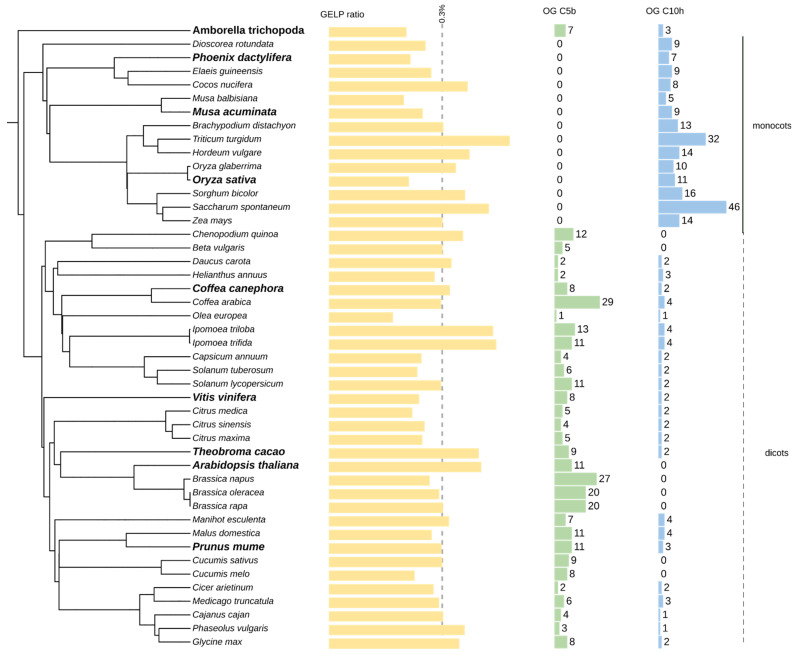
GELP gene distribution in 46 plant species. The species selected for the curated data set are in bold. The GELP bar char indicates the percentage of GELP genes in each species (mean = 0.32%). Other bar charts represent the number of genes in selected orthogroups (OG).

**Figure 4 ijms-23-12114-f004:**
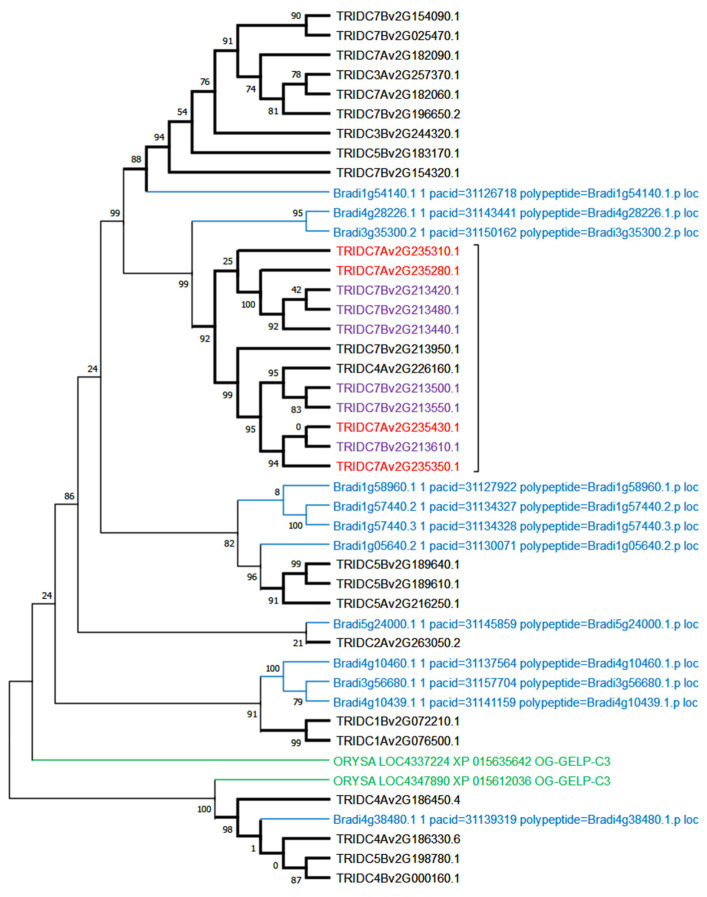
OG-GELP-C3 member amplification in *T. durum*. The phylogenetic tree was constructed with sequences of *T. durum*, *B. dystachion* and *O. sativa*.

**Figure 5 ijms-23-12114-f005:**
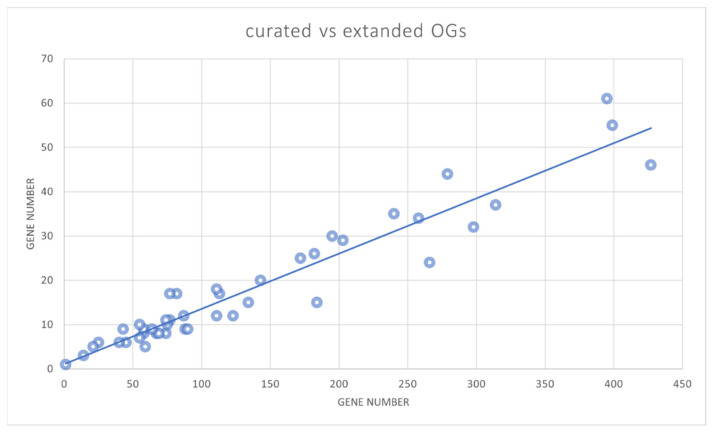
Correlation between the number of GELP members in the curated sequence set (nine species) (*Y* axis) and the member number in the 46 species (*X* axis).

**Table 1 ijms-23-12114-t001:** Iterated phylogenetic analyses.

Step	Sequence Number	Positions	Divergent Cluster (aLRT Support)	Sequences in Cluster
1	248 (66-81-101)	102	Cluster 1 (99)	14 (2-6-6)
2	234 (64-75-95)	113	Cluster 2 (100)	12 (3-5-4)
3	222 (61-70-91)	135	Cluster 3 (100)	9 (1-2-6)
4	213 (60-68-85)	126	Cluster 4 (96)	65 (17-26-22)
5	148 (43-42-63)	179	Cluster 5 (100)	20 (10-2-8)
6	128 (33-40-55)	195	Cluster 6 (98)	10 (4-3-3)
7	118 (29-37-52)	204	Cluster 7 (94)	36 (8-6-22)
8	82 (21-31-30)	253	Cluster 8 (90)	33 (6-13-14)
			Cluster 9 (100)	9 (2-3-4)
			Cluster 10 (97)	40 (13-15-12)

For each step, the number of sequences analyzed (*A. trichopoda*-*P. dactylifera*-*V. vinifera*), number of aligned positions retained for phylogeny inference after alignment cleaning, the name assigned to the divergent cluster (reported in Appendix A), whose sequences were removed in the following analyses (aLRT support of the cluster in the tree), and number of sequences in the divergent cluster are noted. GELP sequences from *M. acuminata*, *O. sativa*, *P. mume*, *T. cacao*, *A. thaliana* and *C. canephora* were assigned to one of the ten clusters using BLASTp if the first five hits belonged to the same cluster. All the query sequences were assigned unequivocally to their respective clusters.

**Table 2 ijms-23-12114-t002:** Overview of cluster analysis to establish orthogroups.

Cluster Name	Sequence Number	Positions	Tree Name
Cluster 1	47 (2-6-5-7-6-4-5-9-3)	231	TreeC1
Cluster 2	37 (3-5-4-4-4-4-4-4-5)	217	TreeC2
Cluster 3	30 (1-2-3-2-6-8-1-5-2)	254	TreeC3
Cluster 4	253 (17-26-35-54-22-23-25-25-27)	169	TreeC4
Sub-cluster 4X	50 (5-2-4-2-10-9-10-4-4)	334	TreeC4X
Sub-cluster 4Y	149 (9-20-24-44-5-6-8-14-19)	220	TreeC4Y
Cluster 5	60 (10-2-0-0-8-12-10-11-7)	284	TreeC5
Cluster 6	35 (4-3-4-3-3-5-6-4-3)	319	TreeC6
Cluster 7	103 (8-6-8-7-22-14-15-15-8)	240	TreeC7
Cluster 8	129 (6-13-12-18-14-13-11-33-9)	262	TreeC8
Cluster 8X	84 (3-6-7-10-10-7-6-29-6)	275	TreeC8X
Cluster 8X1	29 (2-1-2-3-3-2-4-8-4)	295	TreeC8X1
Cluster 8X2	55 (1-5-5-7-7-5-2-21-2)	279	TreeC8X2
Cluster 9	32 (2-3-4-3-4-3-3-5-5)	312	TreeC9
Cluster 10	129 (13-15-20-24-12-15-11-9-10)	213	TreeC10
Cluster 10X	33 (1-4-5-5-4-3-3-4-4)	334	TreeC10X
Cluster 10Y	37 (3-6-10-10-2-3-2-0-1)	249	TreeC10Y

Cluster name, sequence number (*A. trichopoda*, *P. dactylifera*, *M. acuminata*, *O. sativa*, *V. vinifera*, *P. mume*, *T. cacao*, *A. thaliana*, *C. canephora*), the number of aligned positions retained for phylogeny after cleaning, and the name of tree can be found in Appendix A.

**Table 3 ijms-23-12114-t003:** Sequence numbers per species assigned to the established GELP OGs. Species names are represented by species codes. * Two or three sequences represent alternative splicing versions where the first exon is chosen from two (or three in *P. mume*) independent regions. Independent exon structures are marked with different colors, including red for genes in OGs derived from Cluster 1, blue for Cluster 2, green for Cluster 3 and black for the remaining clusters. Exons 2 and 3, which have different junctions in sequences included in the OGs derived from Cluster 4 and Clusters 5–10, are differentiated by a prime symbol (i.e., (2′) and (3′)). In dicots and *O. sativa,* the structures are (1) (2′ + 3′ + 4), (5). In the *A. trichopoda* exon, (3′) is not split into two parts (1), (2′), (3′), (4), (5)).

Orthogroup Name	AMBTC	Dicots and Mono Cots	PHODA	MUSAC	ORYSA	VITVI	PRUMU	THECC	ARATH	COFCA	Exon Structure
OG-GELP-C1a	1	15	4	3	3	1	1	1	1	1	(1), (2)
OG-GELP-C1b	1	30	2	2	4	5	3	4	8	2	(1), (2)
OG-GELP-C2a	1	15	3	1	1	2	2	2	2	2	(1), (2), (3), (4), (5), (6)
OG-GELP-C2b	1	8	1	1	1	1	1	1	1	1	(1), (2), (3), (4), (5), (6)
OG-GELP-C2c	1	11	1	2	2	1	1	1	1	2	(1), (2+3), (4), (5), (6)
OG-GELP-C3	1	29	2	3	2	6	8	1	5	2	(1), (2), (3), (4), (5)
OG-GELP-C4a	1	17	1	2	1	2	4 *	3 *	3 *	1	(1), (2), (3), (4), (5)
OG-GELP-C4b	1	26	2	4	6	4	3	3	3	1	(1), (2), (3), (4), (5)
OG-GELP-C4c	1	8	1	1	1	1	1	1	1	1	(1), (2), (3), (4), (5)
OG-GELP-C4d	1	9	1	1	1	2	1	1	1	1	(1+2+3), (4+5)
OG-GELP-C4e	1	25	2	4	2	6	5	3	2	1	(1), (2), (3), (4), (5)
OG-GELP-C4f	4	20	-	-	-	4	4	7	2	3	(1), (2), (3), (4), (5)
OG-GELP-C4g	8	24	2	1	4	1	1	4	1	10	(1), (2), (3), (4), (5)
OG-GELP-C4h		61	9	12	16	1	3	2	11	7	(1), (2), (3), (4), (5)
OG-GELP-C4i		46	8	10	23	1	1	1	1	1	(1), (2), (3), (4), (5)
OG-GELP-C5a	2	6	2	-	-	1	1	1	-	1	(1), (2′), (3′), (4), (5)
OG-GELP-C5b	8	44	-	-	-	7	11	9	11	6	(1), (2′), (3′), (4), (5)
OG-GELP-C6a	2	12	1	1	1	1	2	4	1	1	(1), (2′), (3′), (4), (5)
OG-GELP-C6b	1	10	1	2	1	1	1	1	2	1	(1), (2′), (3′), (4), (5)
OG-GELP-C6c	1	9	1	1	1	1	2	1	1	1	(1), (2′), (3′), (4), (5)
OG-GELP-C7a	1	8	1	1	1	1	1	1	1	1	(1), (2′), (3′), (4), (5)
OG-GELP-C7b	1	37	2	2	3	8	7	4	8	3	(1), (2′), (3′), (4), (5)
OG-GELP-C7c	1	9	1	1	2	1	1	1	1	1	(1), (2′), (3′), (4), (5)
OG-GELP-C7d	1	11	1	2	-	2	1	2	2	1	(1), (2′), (3′), (4), (5)
OG-GELP-C7e	1	3	-	1	-	1	-	-	1	-	(1), (2′), (3′), (4), (5)
OG-GELP-C7f	1	17	-	-	-	7	1	6	2	1	(1), (2′), (3′), (4), (5)
OG-GELP-C7g	1	9	1	1	1	1	3	1	-	1	(1+2′), (3′), (4), (5)
OG-GELP-C7h	1	1	-	-	-	1	-	-	-	-	(1), (2′), (3′), (4), (5)
OG-GELP-C8a	2	7	1	-	1	1	1	1	1	1	(1), (2′), (3′), (4), (5)
OG-GELP-C8b	1	35	6	5	7	3	5	4	3	2	(1+2′), (3′+4), (5)
OG-GELP-C8c	1	5	-	-	-	1	-	1	1	2	(1), (2′), (3′), (4), (5)
OG-GELP-C8d	1	6	1	1	2	-	1	-	1	-	(1+2′+3′), (4), (5)
OG-GELP-C8e		17	-	1	1	2	1	3	7	2	(1+2′+3′), (4), (5)
OG-GELP-C8f	1	55	5	5	7	7	5	2	21	2	(1), (2′), (3′), (4), (5)
OG-GELP-C9a	1	12	1	2	2	2	1	1	1	2	(1), (2′), (3′), (4), (5)
OG-GELP-C9b	1	18	2	2	1	2	2	2	4	3	(1), (2′), (3′), (4), (5)
OG-GELP-C10a	1	6	1	-	2	1	1	1	-	-	(1), (2′+3′), (4), (5)^a^
OG-GELP-C10b	2	10	1	1	1	1	3	1	1	1	(1), (2′), (3′), (4), (5)
OG-GELP-C10c	1	5	-	-	-	1	1	1	1	1	(1), (2′), (3′+4), (5)
OG-GELP-C10d	3	8	1	1	1	1	1	1	1	1	(1), (2′), (3′), (4), (5)
OG-GELP-C10e	1	9	1	1	2	1	1	1	1	1	(1), (2′), (3′a), (3′b), (4), (5)^b^
OG-GELP-C10f	1	12	1	2	3	1	2	1	1	1	(1), (2′), (3′), (4), (5)
OG-GELP-C10g	1	32	4	5	5	4	3	3	4	4	(1), (2′), (3′), (4), (5)
OG-GELP-C10h	3	34	6	10	10	2	3	2	-	1	(1), (2′), (3′), (4), (5)
Total	66	791	81	95	122	101	101	91	121	78	

**Table 4 ijms-23-12114-t004:** Classification of published GELP genes with functional characterization. The protein sequences of species not sampled for the establishment of OG are in Appendix A.

Gene Name	Gene Function	Species	Classification	References
*Enod8*	Nodule specific	*Medicago truncatula*	OG-GELP-C4e	[23]
*AChE*	Acetylcholinesterase	*Zea mays*	OG-GELP-C4e	[24]
*iEP4*	In carrot suspension cells	*Daucus carota*	OG-GELP-C4e	[25]
*GhGLIP*	Seed growth	*Gossypium hirsutum*	OG-GELP-C4f	[26]
*AChE*	Acetylcholinesterase	*Macroptilium atropurpureum*	OG-GELP-C4f	[27]
*AChE*	Acetylcholinesterase	*Salicornia europaea*	OG-GELP-C4f	[28]
*AAE*	Acetylajmalan esterase	*Rauvolfia serpentina*	OG-GELP-C4g	[29]
*BnLIP2*	Seed germination	*Brassica napus*	OG-GELP-C4h	[30]
*GER1*	Light and jasmonate-induced gene	*Oryza sativa*	OG-GELP-C4i	[31]
*Amgdsh1*	Herbicide activation	*Alopecurus myosuroides*	OG-GELP-C4i	[32]
*Br-sil1*	Salicylic acid- and pathogen-induced	*Brassica rapa*	OG-GELP-C5b	[33]
*TcGLIP*	Pyrethrin synthesis	*Tanacetum cinerariifolium*	OG-GELP-C5b	[34]
*BrGGL7*	Pollen development	*Brassica rapa*	OG-GELP-C7a	[12]
*CaGL1*	Wound stress resistance	*Capsicum annuum*	OG-GELP-C7b	[35]
*CaGLIP1*	Disease susceptibility and abiotic stress tolerance	*Capsicum annuum*	OG-GELP-C7b	[36]
*SaGLIP8*	Cadmium tolerance	*Sedum alfredii*	OG-GELP-C7b	[7]
*JNP1*	Nectar lipid hydrolization	*Jacaranda mimosifolia*	OG-GELP-C7b	[37]
*BrEXL6*	Pollen development	*Brassica rapa*	OG-GELP-C8f	[38]
*EXL4*	Pollen development	*Arabidopsis*	OG-GELP-C8f	[39]
*Xat*	Esterification of lutein	*Triticum aestivum*	OG-GELP-C9a	[40]
*ZmMs30*	Male fertility	*Zea mais*	OG-GELP-C10a	[41]
*OsGELP110/OsGELP115*	Male fertility	*Oryza sativa*	OG-GELP-C10a	[42]
*Rms2*	Male fertility	*Oryza sativa*	OG-GELP-C10b	[14]
*Gelp77*	Male fertility	*Arabidopsis thaliana*	OG-GELP-C10b	[13]
*AgaSGNH*	Leaf epidermis hydrolase	*Agave americana*	OG-GELP-C10g	[43]
*AtLTL1*	Salt tolerance	*Arabidopsis thaliana*	OG-GELP-C10g	[44]
*CUS1*	Cutin synthesis	*Solanum lycopersicum*	OG-GELP-C10g	[45]
*TaGELP073*	Anther and pollen development	*Triticum aestivum*	OG-GELP-C10h	[11]

## Data Availability

The data presented in the current study are available in the article and in the associated Appendix A. The curated set with HMM profiles and a script to expand the current classification are provided at https://github.com/guignonv/search_fasta_hmm (accessed on 6 October 2022).

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
