# Peer review of "Genome-Wide Classification and Phylogenetic Analyses of the GDSL-Type Esterase/Lipase (GELP) Family in Flowering Plants"

_ijms, 2022, doi:10.3390/ijms232012114_

Round 1
Reviewer 1 Report
GDSL-type esterase/lipase (GELP) enzymes are important factors for plant development, and the investigation of GELPs bring much information for researchers working in this field. The authors analyzed representative angiosperm genomes and did much analysis in this study. However, I have some suggestions before the publication of the manuscript.
1. Line 75 to 78, this part should be in the Materials and methods section.
2. There are many errors in sentence punctuations, such as comma, semi- column, etc. The authors need to carefully check and make corrections throughout the manuscript to improve the English writing style for a scientific report.
3. What about the expression patterns of GELP genes in representative genomes? Please try to discuss.
4. Is the gene structure of GELP genes in the same clusters similar among these species? Please provide some information about the gene structures of GELP genes.
Author Response
GDSL-type esterase/lipase (GELP) enzymes are important factors for plant development, and the investigation of GELPs bring much information for researchers working in this field. The authors analyzed representative angiosperm genomes and did much analysis in this study. However, I have some suggestions before the publication of the manuscript.
- Line 75 to 78, this part should be in the Materials and methods section.
Answer: This part is now included in Materials and methods (4.2 Phylogenetic analyses). Moreover, the sentence Line 75 needed to be adjusted due to deletion and was rephrased as “The first global GELP phylogeny that included A. trichopoda, P. dactylifera, and V. vinifera (66, 81 and 101, respectively) resulted in 102 positions after alignment filtering (Table 1), approximately accounting for only one-fourth of the GELP sequence average size.”
- There are many errors in sentence punctuations, such as comma, semi- column, etc. The authors need to carefully check and make corrections throughout the manuscript to improve the English writing style for a scientific report.
Answer: Sorry for the inconvenience. The manuscript has been reviewed by a native English speaker and errors must have been corrected and some sentences made clearer.
- What about the expression patterns of GELP genes in representative genomes? Please try to discuss.
Answer: GELP gene expression pattern can be of interest. However, compared to most of the studies that focused on a single plant genome and included their gene expression analyses, we are dealing in this study with a much larger set of species. Getting homogenous data for the same tissues is challenging (even in NCBI Geo patterns) even for the reduce sample of representative genomes, and prone to make erroneous conclusions. Identify whole transcriptome datasets in comparable tissues to be processed, filtered to GELP was considered not in the scope of the study.
- Is the gene structure of GELP genes in the same clusters similar among these species? Please provide some information about the gene structures of GELP genes.
Answer: The exon/intron structure of the GELP genes was analyzed and discussed in section 2.3. Gene structure of GELP genes and in Table 3. We observed that globally within OG the structure was consistent with a few exceptions and after curation of erroneous annotations (chimeric sequences, missing exons).
At the protein domain level, we performed InterProScan analyses for the 855 proteins composing representative genomes. The are some differences between the most differentiated clusters but within OGs, the protein domain structure was conserved between species. We have added a paragraph in the result section for this analysis as well as a new supplementary file with type and percentage of functional domains by orthogroups.
Reviewer 2 Report
Overall, this research article represents an interesting investigation on “Genome-Wide classification and phylogenetic analyses of the GDSL-type esterase/lipase (GELP) family in flowering Plants”. Abstract is logical providing the concise summary of the findings, The introduction provides sufficient background, the methods are generally appropriate for the experiments conducted. The analysis and results presented in figures seem logical while interpretation is supported by results. Moreover, the results are clearly described making the manuscript understandable for readers. In order to improve the present study, some essential modifications have to be fixed before it proceeds, and decisive action can be taken. In addition, the study needs some editing on some minor grammatical issues. All the comments and remarks are given below.
In introduction, line 30, (Upton and Buckley 1995), better to cite it according to journal format.
In line 110-111, authors have stated that “To establish monocot/dicot orthogroups (OGs) for the whole GELP family, ten clusters, including the GELP sequences from A. trichopoda, three monocots, and five dicots, were submitted to cluster-specific phylogenetic analysis”, while in line 309 & 486 the authors have stated that, “A. trichopoda genes were not included in the OGs”, why? Please explain.
Why the A. trichopoda genes were not considered for HMM pattern definition? Although they are included in clustering and other analysis.
The quality of Figure 2. Should be improved.
In figure 3. “GELP gene distribution in 46 plant species. The species selected for the curated data set are in bold”. Unable to differentiate between bold and normal fonts, either increase the font size or change the font color so that the species selected for the curated data set can be easily differentiated.
Author Response
Overall, this research article represents an interesting investigation on “Genome-Wide classification and phylogenetic analyses of the GDSL-type esterase/lipase (GELP) family in flowering Plants”. Abstract is logical providing the concise summary of the findings, The introduction provides sufficient background, the methods are generally appropriate for the experiments conducted. The analysis and results presented in figures seem logical while interpretation is supported by results. Moreover, the results are clearly described making the manuscript understandable for readers. In order to improve the present study, some essential modifications have to be fixed before it proceeds, and decisive action can be taken. In addition, the study needs some editing on some minor grammatical issues. All the comments and remarks are given below.
Answer: Thank you for the positive comments and the time to review our manuscript in order to suggest useful improvements.
In introduction, line 30, (Upton and Buckley 1995), better to cite it according to journal format.
Answer: This is an oversight when converting the references. This is now fixed. Thank you for reporting.
In line 110-111, authors have stated that “To establish monocot/dicot orthogroups (OGs) for the whole GELP family, ten clusters, including the GELP sequences from A. trichopoda, three monocots, and five dicots, were submitted to cluster-specific phylogenetic analysis”, while in line 309 & 486 the authors have stated that, “A. trichopoda genes were not included in the OGs”, why? Please explain.
Answer: A. trichopoda genes were used as an outgroup because it has a basal position to monocot/dicot lineage split. This is a way to clearly delineate orthogroups based on the expected species tree. For most of the OGs, one A. trichopoda sequence (or one cluster of A. trichopoda sequences) confirmed the correctness of the OG definition.
As stated in paragraph 4.3 (Materials and Methods), our orthogroups definition is relative to the monocot and dicot last common ancestor. One of the main reasons for this choice is that the monocot/dicot clade includes almost all the crops for which the functional transfer is relevant to breeding.
To make it more explicit before the Materials and Methods (which is at the end of the article), we modified the following sentence in the introduction line 65 “In this study, we performed a semi-automatic curation of genes of the complex and large gene family of GELP using multi-step phylogenetic analyses to establish orthogroups, relative to the ancestral split between monocot and dicots.”
Why the A. trichopoda genes were not considered for HMM pattern definition? Although they are included in clustering and other analysis.
Answer: A. trichopoda genes were not considered for HMM pattern for reason explained above. It was to be consistent with our orthogroups definition.
The quality of Figure 2. Should be improved.
Answer: the figure 2 was fully redrawn in circular mode with improved design and resolution.
In figure 3. “GELP gene distribution in 46 plant species. The species selected for the curated data set are in bold”. Unable to differentiate between bold and normal fonts, either increase the font size or change the font color so that the species selected for the curated data set can be easily differentiated.
Answer: the figure 3 was improved by increasing the font size (in case it is printed in black and white).